# Differences in Work and Commuting Accidents between Employees and Students at Higher Education Institutions in Rhineland-Palatinate, Germany, from December 2014 to December 2019

**DOI:** 10.3390/ijerph20032462

**Published:** 2023-01-30

**Authors:** Lea Ursula van der Staay, Clemens Koestner, Pavel Dietz

**Affiliations:** Institute of Occupational, Social and Environmental Medicine, University Medical Center of the Johannes Gutenberg University Mainz, 55131 Mainz, Germany

**Keywords:** accident, injury, prevalence, traffic, student

## Abstract

Accidents are one of the most important public health concerns because of their high prevalence and considerable health outcomes. Although higher education institutions (HEIs) play an important role in health promotion and disease prevention, accidents are rarely investigated in this setting. Therefore, the aim of the present study was to address this gap by analyzing the frequency and characteristics of employee and student accidents at HEIs in Rhineland-Palatinate, Germany. A dataset of all accidents that happened to employees and students at HEIs in Rhineland-Palatinate from December 2014 to December 2019 and the characteristics of these accidents was provided by the responsible statutory accident insurance (Accident Insurance Fund of Rhineland-Palatinate). Modified thousand-men quotas (the rate of injuries per 1000 people) were calculated to investigate the differences in frequencies and characteristics of accidents between employees and students, as well as between institutions. A total of 3810 accidents (n = 1326; 34.8% work and n = 2484; 65.2% commuting) were reported, of which 426 involved employees and 3384 involved students. The frequency and characteristics of the accidents varied between employees and students, as well as between institutions. Sports programs at HEIs for example imply high risks for unintentional injuries especially for students (as they make up the majority of participants). Other main findings are that medical students, as well as students of subjects including laboratory work, are at a higher risk to experience study-related accidents whereas employees seem to be at a higher risk when working in a technical field. The results call for the development of accident prevention concepts at HEIs and the implementation of interventions in respective institutions and target groups.

## 1. Introduction

### 1.1. Definition and Prevalence of Accidents

In German law, the term “accident” is defined as a temporally limited incident that externally affects the body and leads to death or damage to one’s health. Accidents are grouped as either work or commuting accidents. A work accident happens as a result of an insured occupation, whereas a commuting accident happens on an insured person’s direct way to or from their place of occupation [1]. It has to be mentioned that, according to this definition, the term work accident does not only refer to accidents that happen to employees but also to accidents that happen to pupils and students, for example. It must be stressed that the event of an accident is never intended.

As the World Health Organization (WHO) states [2], injuries take the lives of 4.4 million people around the world each year and thus account for nearly 8% of all deaths. For people aged between five and 29 years, three of the top five causes of death are injury related, namely, road traffic injuries, homicide, and suicide. Millions of people each year experience non-fatal injuries, which must be treated professionally and often result in temporary or even permanent disability, with the need for long-term physical and mental health care and rehabilitation. For example, there has been a significant increase in road traffic injuries in the African region since 2000, with an almost 50% increase in healthy life years lost [3].

Among other data related to peoples’ health status, health determinants, health care, and accident and injury statistics for the European Union (EU) are collected by the European Health Interview Survey (EHIS) [4]. The data include all EU member states, and the survey is conducted every five years. Accident and injury statistics [5] focus on four aspects: deaths from accidents, deaths from assault, the extent of accidents, and health care for injuries. Accidents accounted for approximately one fifth or more of all deaths, peaking at 36% for people aged between 20 and 24 years. The results of the EHIS in Germany for the years 2014 and 2015 showed that in the past 12 months, 8.6% of all adults in Germany suffered injuries from accidents that required medical treatment. Young adults aged between 18 and 29 years were often involved in accidents that required medical attention, presenting an 18.1% accident injury rate over the past 12 months among men and a 9.8% accident injury rate among women. Further gender-related differences in the prevalence of occupational accidents were reported. In numerous male-dominated professions, such as the building sector, the accident risk was higher [2].

A summary of data on German accidents, such as those that occurred (i) during leisure time (3.9 million), (ii) at home (3.2 million), (iii) at school (1.3 million), (iv) at work (1 million), and (v) in traffic (0.4 million), showed that around 9.37 million accidents happened in Germany in 2015. Almost 24,600 of these resulted in death [6].

### 1.2. Public Health Relevance of Accidents

Because of their high prevalence and considerable health outcomes, accidents are one of the most important public health concerns [7]. Injuries are common causes of mortality and permanent disability in young people living in developed countries [8]. The WHO states that most intentional and unintentional injuries can be prevented, highlighting that public health, as a discipline, plays an important role in health promotion and accident prevention. Injuries can be considered predictable, and there is elaborate scientific evidence of what works to prevent them and address their consequences in various settings [2]. An example showing that even small investments of money can have major impacts on prevention is mentioned in a WHO report about injuries and violence. It was shown that, for example, every USD 1 invested in smoke detectors saves USD 65 in medical costs, productivity loss, and property loss; in child restraints and bicycle helmets, it saves USD 29; and in in-home visitation, it saves USD 6 [3]. Furthermore, the social benefits of injuries prevented through home modifications that prevent falls are estimated to be at least six times the cost of intervention [9].

### 1.3. Higher Education Institutions as Settings of Public Health Relevance

Institutions of higher education are settings of public health relevance. Extraordinary situations can be observed there, where two life-worlds exist: those of work and those of study. The interaction of these two groups is highly essential to our future society. The Ottawa Charter for Health Promotion in 1986 emphasized the importance of a supportive environment to enable people to increase control over and improve their health. Therefore, aside from other institutions, HEIs are tasked to create a supportive environment that enables individuals to access information and acquire skills so that they can make healthy life choices [10]. Based on the Ottawa Charter, the importance of higher education and its local and global influence on the development of individuals, societies, and cultures was stated in the Okanagan Charter in 2015 [11]. It highlights the interdependence of the well-being of people, places, and our planet and concludes that HEIs are key parts of systemic health promotion.

Among approximately 2.95 million students enrolled at German HEIs (winter term 2020–2021) [12], a considerable discrepancy has been observed in this collective’s accident data. Attempts have been made to raise awareness of study-related accidents, but only a limited number of studies have dealt with this particular issue. Only a few studies from other European countries describing accident rates among higher education students can be found, and these typically use survey techniques. One of these surveys collected information on students at three institutions in the UK in 2005 via a postal questionnaire. The survey found that 18% (222 out of 1208 students) reported at least one injury requiring medical attention in the last year. Injuries caused by physical activity were more likely to be disabling than others were [8]. Another example is a survey-based analysis of traumatic injuries among 617 randomly selected third-year higher education students in Finland, in which 28.7% (177 out of 617) of the students reported 281 accidents in total, and 323 separate injuries required medical treatment; only 0.5% of the accidents were classified as study-related [13]. Another example is a multicenter cross-sectional health survey conducted at 16 HEIs in North Rhine-Westphalia, Germany, in 2006 and 2007. It showed that 8.8% (252 out of 2855) of the participants experienced an accident in the context of their studies. Approximately 60% of the accidents occurred during study-related sports activities, and 25% were commuting accidents that happened on the students’ way to or from HEIs [14]. 

Regarding the identification of groups with a higher risk of encountering accidents during their studies, cross-sectional anonymous surveys were carried out among medical undergraduate students at the Charité Berlin in 2009 and 2019 to assess whether they experienced needlestick and sharps injuries. In addition, an analysis of all needlestick and sharps injuries reported to the accident insurer from 2007 to 2010 was performed in one institution. This university is one of Germany’s major medical faculties and was educating more than 7000 undergraduate medical students in 2010. The survey was completed by 2131 students in total. The results showed an injury rate of 21.4% per year (mean value), with a rate of underreporting of more than 50%. The study concluded that undergraduate medical students are at a high risk of suffering from needlestick and sharps injuries. It emphasized the importance of using safe instruments and following university instructions to prevent such injuries [15]. A total of 101,712 medical students in winter term 2020–2021 [16], accounting for about 1/29 of the 2.95 million students enrolled at German HEIs, are an important group to examine; however, the remaining 28/29 of these students must be investigated to create effective programs for health promotion and the prevention of study-related injuries.

There is a more recent study in one institution of higher education, the Johannes Gutenberg University Mainz (JGU), which is the largest university in Rhineland-Palatine, Germany. The study is the only one to date that uses objective accident data. These data were provided by the Accident Insurance Fund of Rhineland-Palatinate and indicated that there were 1285 study-related accidents among students at the JGU from December 2012 to December 2018. The study also revealed that students in the medical faculty, chemistry faculty, pharmaceutical sciences, and geography and geosciences, as well as students who participate in study-related sports activities, administered by the General University Sports Department had a significantly higher proportion of work accidents in the context of their studies [17]. 

The other group on which this study focuses is employees at HEIs, who are subdivided into two. The first group consists of science support staff working in the fields of management and technical support. The second consists of employees in the field of science who are in teaching positions [18]. It is important to differentiate between these two types, as their occupational tasks involve different accident risks.

While there has been no study that focused on science support staff and their injury rates, a few studies have examined groups of schoolteachers. This might be comparable to the group of employees at HEIs occupying teaching positions. An analysis of accident registration forms, for example, described the workplace accidents involving teaching staff in Rhineland-Palatinate during the academic years of 2012–2013 and 2013–2014 [19]. During these years, 847 workplace accidents were reported, and women and younger teachers were identified as high-risk groups. Another study in Spain analyzing all teachers’ accidents recorded from 2003 to 2018 (a total of 136,702) showed that the most important factors contributing to a higher risk of suffering a neck injury in the context of commuting accidents by car were having female sex, being under 45 years of age, being in the first month of teaching, and belonging to a public school without risk assessment [20]. However, conclusions on science employees at HEIs cannot be drawn from this information. The examples presented indicate the great heterogeneity in the assessment of study-related accidents at HEIs and in the classification of the individuals involved (whether students or employees). The identification of groups with a higher risk of experiencing accidents throughout their studies has rarely been pursued, either internationally or nationally. Therefore, access to the accident rates of a larger population, such as the German federal state of Rhineland-Palatinate, might be pioneering to determine high-risk groups at HEIs and can be a valid foundation for fostering better health promotion activities at campuses of higher education. Furthermore, given the heterogeneity in assessing accident rates, this study used the modified thousand-men quota (mTMQ) [19] to calculate the relative overall frequency of accidents and enable comparability in the assessment of information.

### 1.4. Aims of the Study and Research Questions 

To address the above-mentioned knowledge gap, this study aimed to: (i) calculate for the first time the relative overall frequency of accidents involving employees and students at HEIs in Rhineland-Palatinate, Germany; (ii) investigate differences in the relative overall frequency of accidents between employees and students; (iii) examine differences in the relative overall frequency of accidents and the distribution of work and commuting accidents between different HEIs; and (iv) explore the differences between employees and students and further sociodemographic factors in terms of the relative frequency of work and commuting accidents.

This information may be useful to determine particular institutions and groups that are at risk, thus helping in the development of health promotion concepts and the implementation of appropriate interventions.

## 2. Materials and Methods

### 2.1. Basic Population

The German state of Rhineland-Palatinate has a great variety of HEIs. Consisting of state-run and private universities, as well as extramural research facilities, the higher education landscape of Rhineland-Palatinate covers a wide range of the technical, scientific, and humanistic spectrum [21]. There are five universities in Rhineland-Palatinate: the JGU (uniting almost all academic disciplines under one roof, including the Mainz University Medical Center, the Mainz Academy of Fine Arts, the Mainz School of Music, and the Faculty of Translation Studies, Linguistics, and Cultural Studies in Germersheim), the Technical University Kaiserslautern, the University Koblenz-Landau (mainly providing teacher training but also including the humanities, cultural studies, and natural and computer sciences in its academic profile), the Trier University (mainly covering social studies), and the German University of Administrative Sciences Speyer. 

In addition, the state has seven universities of applied sciences: the Bingen Technical University of Applied Sciences, offering degree courses in engineering, information and communications technology, and life sciences; the University of Applied Sciences Kaiserslautern, mainly providing courses in the engineering and science fields; the University of Applied Sciences Koblenz, which has a variety of faculties in the social and technical or administrative departments; the Ludwigshafen University of Business and Society, covering a range of degree programs in the fields of business management, social work, and healthcare; and the University of Applied Sciences Mainz, with its network of three schools of design, engineering, and business. Finally, the Trier University of Applied Sciences, with its three campuses offering diverse educational programs, such as engineering, economics, law, and design environmental studies, and the University of Applied Sciences Worms, with both business administration and technical bachelor’s and master’s study programs, are also universities of applied sciences. Additionally, there is the Catholic University of Applied Sciences Mainz, which offers courses in the fields of social work and social sciences, practical theology, and healthcare and nursing. 

### 2.2. Dataset

The dataset for this work was provided by the Accident Insurance Fund of Rhineland-Palatinate (Unfallkasse Rheinland-Pfalz). Together with 15 other local accident insurance funds, it makes up the German statutory accident insurance. The German social security system is composed of four obligatory insurances. One of these is the German Statutory Accident Insurance (Deutsche Gesetzliche Unfallversicherung), which ensures that employees, children, and students are insured during their activities at the workplace and their places of education. In the event of a commuting accident or an accident at work resulting in the inability to work, treatment by a doctor, or even death, the institution is required to report the accident to its relevant local accident insurance fund [22]. As the current study focused on the work and commuting accidents of employees and students at HEIs in Rhineland-Palatinate, the Accident Insurance Fund is the only insurance fund considered. The dataset provides information about all reported accidents in such institutions in Rhineland-Palatinate from December 2014 to December 2019. The dataset did not include any personally identifiable information (e.g., name, birthday, or exact date of the accident) to protect the privacy of the insured persons. The following variables were analyzed in this study: (i) gender (f/m), (ii) age, (iii) category of persons (students/employees), (iv) institution, (v) type of accident (work or commuting), (vi) general reason for the accident (e.g., injured by falling or by mechanical impact), and (vii) specific reason for the accident (e.g., failed rotation or colliding with someone). Formal approval by the local ethical committee was not required, given that the anonymous dataset was provided by the Accident Insurance Fund of Rhineland-Palatinate.

### 2.3. Statistical Analysis

The results presented in this paper are mostly descriptive. The total number of accidents and their types (work or commuting accidents) among employees and students at HEIs in Rhineland-Palatinate from December 2014 to December 2019 are presented. The general and specific reasons for the accidents are also differentiated. These characteristics are presented as absolute numbers and relative percentages. To test categorical variables for statistically significant differences, Chi² tests (*p*-values were Bonferroni adjusted if cross tables exceeded a 2 × 2 size, by calculating alpha-values as *p* = 0.05/number of tests) were calculated. Analyses of variance (ANOVA) were used to assess differences for continuous variables in specific subgroups. Requirements (Chi² Test: expected frequencies for cells > 1, expected frequencies > 5 for more than 80% of the cells. ANOVA: normality, outliers, homoscedasticity) for the conduction of the respective analysis methods were tested and met. In addition, the mTMQ [19] was calculated considering the mean basic population of each institution. To assess the number of employees and students at each institution, the study used the data collected by the Federal Office of Statistics Rhineland-Palatinate [23,24]. Statistical reports from 2014 to 2019 present the detailed data on employees and enrolled students for each HEI in the state, which are summarized in Appendix A (Appendix A). If there was missing information, the responsible staff of the certain institution were contacted, and the missing numbers were obtained. The mean basic population throughout the five-year period was calculated to determine the mTMQ for the entire period from December 2014 to December 2019. Because the years of 2014, 2015, 2019 and 2020 have not been completely reported, the sum of the number of winter terms in academic years 2014–2015 and 2019–2020 was divided by two to identify the mean number of students enrolled throughout the total five years’ worth of data. In this way, the mean value of the two winter terms that were not recorded completely in the dataset provided by the Accident Insurance Fund of Rhineland-Palatinate was identified. There were records of only one month in 2014 and 11 months in 2019. The results were added up with the number of enrolled students from the winter terms in academic years 2015–2016, 2016–2017, 2017–2018, and 2018–2019. In the next step, this sum was divided by five (because there was a total of five years considered in the dataset). The results represent the mean number of enrolled students from December 2014 to December 2019. The mean number of employees from December 2014 to December 2019 was calculated using the same method, again by first determining the mean value for the years of 2014 and 2019. This number was then added up with the numbers of employees in 2015, 2016, 2017, and 2018 and then divided by five. 

The mTMQ describes the rate of injuries per 1000 employees or 1000 students from December 2014 to December 2019. It was calculated using the following formula: mTMQ = (a · 1000)/n.

In sum, the total number of accidents (denoted as a) during this period was added up, multiplied by 1000, and then divided by the average number of employees or students at the institution during the five-year period (denoted as n). 

## 3. Results

The total number of reported study-related accidents from December 2014 to December 2019 was 3810. Employees were involved in 426 of these, and the remaining 3384 accidents involved higher education students. As shown in Table 1, a total of 1326 (34.8%) accidents were commuting related, whereas 2484 (65.2%) were work accidents. For employees, 128 (30.1%) accidents were commuting related, whereas 298 (69.9%) were work accidents. For higher education students, 1198 (35.4%) accidents were commuting related, whereas 2186 (64.6%) were work accidents. The results of the Chi² Test showed that there was a significant difference between the type of accident (commuting vs. work) and the category of person (employees vs. students), χ² (1, *N* = 3810) = 4.78, *p* = 0.03. Significant differences were found too between the types of accident and gender (female vs. male), χ² (1, *N* = 3810) = 95.80, *p* < 0.001. The results of an ANOVA revealed no significant differences in the mean values of age at the time of accident (*N* for the ANOVA was slightly reduced due to missing values) when the type of accident was used to divide into subgroups: F(1, 3806) = 1.29, *p* = 0.26.

The calculated modified thousand-men quotas (mTMQs) for the different HEIs are shown in Figure 1 and Appendix A. It can be seen that the numbers differ between the different HEIs, as well as between employees and students. The calculations enabled the identification of particular institutions with higher risks of accidents for employees and students. The obtained results revealed the following two HEIs to have the highest accident rate of either employees or students. It needs to be mentioned that the dataset provided can only help us observe and suggest where the focus on risk groups should be and a final claim cannot be made. At the German University of Administrative Sciences Speyer, there were 11.79 injuries per 1000 employees from December 2014 to December 2019. Regarding students’ injury rates, the University of Koblenz-Landau had the highest rate, with 8.2 injuries per 1000 students within that period of time. Furthermore, we examined the accident characteristics of the three institutions with the highest mTMQ (Figure 1). These were the German University of Administrative Sciences Speyer, the University of Applied Sciences Trier, and the University of Applied Sciences Kaiserslautern for employee accidents and the University of Koblenz-Landau, Technical University Kaiserslautern, and the JGU for student accidents. 

### 3.1. University Students

Referring to the characteristics of students’ accidents in Table 2, Table 3 and Table 4, a major result is the high incidence of injuries due to falling in the context of work accidents. 

At the University of Koblenz-Landau, 665 accidents were reported by students from December 2014 to December 2019. For example, work accidents accounted for 61% (n = 403) of these, and the main general reason (n = 213; 52.9%) was injuries by falling. Two hundred sixty-two (39.0%) of the students’ accidents were reported as commuting related, mostly with the general reason of injuries by a bump or hit (n = 163; 62.2%). In Table 2 you can find more information about the specific accident data of the University of Koblenz-Landau.

At Technical University Kaiserslautern, whose data are presented in Table 3, a total of 406 accidents were reported by students from December 2014 to December 2019. Work accidents comprised 79% (n = 321) of these, and the main general reason (n = 138; 43.0%) was again injuries by falling. Of the students’ accidents, 85 (21.0%) were reported as commuting related, mostly with the general reason of injuries by a bump or hit (n = 43; 50.6%). Further information is documented in Table 3.

At the JGU, a total of 771 accidents were reported by students from December 2014 to December 2019. The data is represented in Table 4 and shows, among other things, that work accidents comprised 85.6% (n = 660) of all accidents reported, and the main reason (n = 279; 42.3%) was injuries by falling. A total of 111 (14.4%) accidents were reported as commuting related, mostly with the general reason of an injury by falling (n = 65; 58.6%). More results are presented in Table 4.

### 3.2. Employees

From the data on employees’ accident characteristics, as summarized in Table 5, 12 accidents were reported by employees at the German University of Administrative Sciences Speyer from December 2014 to December 2019. Work accidents accounted for 58.3% (n = 7) of these, and the main general reason (n = 5; 71.4%) was injuries by falling. 

At the University of Applied Sciences Trier, a total of 18 accidents were reported by employees from December 2014 to December 2019. Work accidents accounted for 72.2% (n = 13) of these, and the main general reason (n = 5; 71.4%) was injuries by a bump or hit (n = 6; 46.2%).

At the University of Applied Sciences Kaiserslautern, a total of 13 accidents were reported by employees from December 2014 to December 2019. Work accidents accounted for 53.8% (n = 7) of these, and the main general reason was being injured by physical, chemical, or biological impact (n = 3; 42.9%). All these cases involved damage through hazardous substances (n = 3; 100%). Another three of the reported work accidents (42.9%) occurred because of a mechanical impact. Of the work accidents at the University of Applied Sciences Kaiserslautern, 14.2% (n = 1) were reported as injured by falling, specifically an employee who twisted their ankle. A total of 46.2% (n = 6) were commuting accidents, and 50.0% (n = 3) of these were classified as injured by falling. Three employees (50.0%) were involved in commuting accidents because of injuries by a bump or hit; more precisely, they collided with someone or something.

A Chi² Test (2 × 13 matrix with Bonferroni adjusted *p*-values to counter effects of type I error accumulation) for type of accident (work vs. commuting) and institutions showed that significant differences in type of accidents were only found at the Johannes Gutenberg University Mainz and the University of Applied Sciences Kaiserslautern.

## 4. Discussion

This study aimed to gain more information about the relatively understudied topic of accidents (injuries) reported among adult employees and students in higher education institutions (HEIs). In particular, this was done by calculating for the first time the relative overall frequency of accidents involving employees and students at HEIs in Rhineland-Palatinate, Germany, to investigate the differences between the two groups. Another aim was to examine the relative overall frequency of accidents and the distribution of work and commuting accidents between different HEIs, as well as between their employees and students. This study used data on accidents provided by the Accident Insurance Fund of Rhineland-Palatinate (Unfallkasse Rheinland-Pfalz), through which we calculated the modified thousand-men quota (mTMQ) for the different HEIs and their groups of employees and students.

### 4.1. Socio-Demographic Insights

In general, we can say that more accidents involving students have been reported than accidents involving employees. However, we believe that this result is only consequential, as the total number of students at the HEIs covered in the present paper is always higher than the total number of employees. 

Furthermore, there were slightly more women than men involved in study-related accidents. The percentage of women at HEIs has increased over the last few years (in 2020, 52.5% of the new students enrolled at German HEIs were women) [25], and the majority of employees are women as well (accounting for 54.1% of the total staff at German HEIs) [25]. Therefore, we could draw the conclusion that it is only consequential to have more accidents reported by women. 

Nevertheless, we could also call into question whether women might be at a greater risk, as our study showed that females had a significantly higher risk to experience commuting accidents. While several studies of unintentional injuries in children and adolescents reveal boys to suffer unintentional injuries more frequently than girls do [26,27], there have been studies that reveal female young adults to be at a higher risk than male ones. An Italian study used data on recognized commuting while walking injuries in the industrial and service sectors recorded by the Italian National Institute for Insurance against Accidents at Work from 2014 to 2018; the study found that during the years examined, commuting while walking injuries occurred three times more often among women, with an increasing incidence over the years and by age [28]. An older study from Spain obtained similar results, analyzing a total of 266,646 traffic-related injuries and 168,129 nontraffic-related injuries from 2006 to 2010. The study also found that commuting-related accident rates were higher among women than men for both traffic- and nontraffic-related injuries [29]. However, as mentioned above, we cannot draw a final conclusion whether women might be at a higher risk or whether it is only a consequence of women making up the majority of students. Therefore, discussing a gender-oriented prevention approach at HEIs might be important because, as our dataset showed, we cannot exclude the possibility of women being at a greater risk of being involved in study-related accidents, especially commuting accidents. In contrast to that, we could also reveal a significant difference of males having a higher risk for work accidents. This might be a consequence of more males than females studying and working in engineering and technical fields which normally includes working with several types of machines. However, as the topic and definition of gender are currently fluid, a concept such as gender-oriented prevention might not be future proof at all. 

Another noteworthy result regarding the socio-demographic variables might be that the age range of employees and students is not as far apart as one might have expected. A reason could be that many students have part-time jobs at the university, such as helping out at the library or doing research for their faculty. However, with a mean age of 37.7 years, the employee group might still cover mostly full-time staff. Regarding the age range of students going up to 55 years, mature students needed to be considered as well. Although they represent a minority in the student group, they might usually have more life and work experiences and more qualifications than adolescents, which might be a possible explanation for this collective to be less at risk of suffering study-related accidents. However, our study could not show a significant association between age and the type of accident. Further investigation needs to be done concerning the issue of mature students. However, as in terms of age, the collective of employees might be more representative for the collective of mature students, certain arguments for their risk to be higher than those of younger might apply as well. One example could be age as a risk factor for falls and falls with serious outcomes on ones’ health (as mentioned in Section 4.2.2). Due to a lack of more data on this particular collective and its small total number in our dataset, a final conclusion cannot be drawn. Furthermore, our results showed that different HEIs seemed to have different accident risks, and employee and student groups did not seem to have equal risks. 

### 4.2. Work Accidents

#### 4.2.1. Students

##### Sports

There were twice as many reported work accidents as commuting accidents in the student group. In particular, the numbers of reported accidents at the University of Koblenz-Landau, Technical University Kaiserslautern, and the JGU were striking. At the University of Koblenz-Landau, injuries by falling accounted for more than half of all reported work accidents; injuries by falling also accounted for the majority of work accidents at Technical University Kaiserslautern (138 of 321 work accidents within the five-year period presented) and the JGU (279 of 660 work accidents within the five-year period presented), respectively. The second leading reason for work accidents at two of the institutions was injuries by a bump or hit (almost a third of all work accidents at the University of Koblenz-Landau and almost one fourth of all work accidents reported at the Technical University Kaiserslautern), which were also the third leading reason for work accidents at the JGU (narrowly beaten by injuries by mechanical impact with 151 injuries versus 148 injuries by bump or hit). 

Regarding the more specific reasons for accidents (i.e., twisting one’s ankle or falling or tripping in the case of injuries by falling and getting hit by something or colliding with someone/something) and based as well on the accident descriptions (included in the dataset provided but not further encoded and thus not further mentioned elsewhere), we can conclude that the majority of these work accidents happened in the context of HEIs’ sports programs. This conclusion is supported by the results of previous studies based on student surveys [13,14]. These insights show that there is an urgent need for prevention programs concerning sports injuries. A national survey in the Netherlands using a computer-assisted telephone survey on sports injuries and sports participation from 2000 to 2005 showed that sports participation was associated with 1.5 million injuries per year. Of these, 50% had to be treated medically. The injuries accounted for two thirds of all direct and indirect costs (EUR 400 million) [30]. 

These results might be helpful as a guide for prevention programs focusing on certain sports and for ensuring that participants have a full understanding of the sport and its dangers. As sports programs at universities are normally provided per semester, students at HEIs are given the chance to try out new and different types of sports. This also implies the risk of rather careless behavior; one might not focus much on a sport’s safety instructions, as opposed to one who has been engaged with the sport for years, as they only intend to practice for half a year. It would be important for the prevention initiatives of HEI sports programs to tackle this problem and create awareness of safety between experienced and non-experienced sportsmen. 

##### Needlestick Injuries

The second leading cause of work accidents at the JGU was injuries by mechanical impact. For the two other HEIs discussed, it was the third leading cause of work accidents. The JGU is the only institution of the three that provides medical education, which apparently implies a higher risk of students suffering from needlestick injuries (NSIs). Previous studies have identified medical students as a group with a higher risk, similar to an anonymous electronic survey among surgical personnel, which showed that 22% of all surveyed medical students had a history of NSIs [31]. 

Comparable results were observed in a study of study-related work and commuting accidents among students at the JGU, revealing that medical students have an increased risk for study-related work accidents. Almost three quarters of all accidents that occurred among medical students were NSIs [17]. Another cross-sectional anonymous survey was carried out among 2131 medical undergraduate students at the Charité Berlin in 2009 and 2019 to assess whether they had experienced NSIs. The study concluded that undergraduate medical students were at a high risk of suffering from NSIs, thus emphasizing the importance of using safe equipment and following safety instructions to prevent NSIs [15]. A cross-sectional study conducted in 2012 and 2013 at Hormozgan University of Medical Sciences (Iran) additionally found that the major mechanism of injury was by damaging vein puncture [32]. All the reported findings of other groups support our results and demonstrate that NSIs are common among medical students.

##### Injuries in a Lab Setting

Injuries by mechanical impact do not occur to medical students only; they also have a high prevalence at HEIs that do not offer medical education. Therefore, we need to identify other student groups at risk. Regarding the educational programs at the University of Koblenz-Landau and Technical University Kaiserslautern (as well as at the JGU), subjects in the fields of chemistry, pharmaceutical sciences, geography, and geosciences are also offered. Evidently, part of the studies in these disciplines takes place in the lab and partially includes working with sharp objects, such as syringes or objects made of glass that are prone to break easily. This might explain the higher prevalence of injuries caused by mechanical impact. 

Furthermore, injuries by physical, chemical, or biological impact are common causes of accidents in the lab as a working environment. The accidents reported at the JGU and Technical University Kaiserslautern support this finding, with the discussed injuries being the fourth most common reason for work accidents. The main reasons for injuries by physical, chemical, or biological impact were damage through hazardous substances and burning oneself. Labs in academic settings were assessed in 1986 as having higher accident rates than others [33,34]. However, the assessment of study-related accidents, in particular, is rather underrepresented. The evaluation of laboratory-related injuries at Iowa State University from 2001 to 2014 revealed that students (in this case, a group comprising graduate assistants and student employees) were the most frequently injured group, accounting for 40.9% of all laboratory-related injuries. In addition, the analysis of the dataset identified lacerations as one of the most prevalent injury types, 74.4% of which were attributed to glassware breaking during lab work [35]. These results support the findings of the current study.

The above results should be a call to take action toward limiting these types of injuries. Increasing safety awareness in this context is important. Taking safety steps before starting to work and experiment, such as using appropriate containers and inspecting glassware to ensure that it is not cracked or broken, can have a major impact on the prevention of lab accidents. Support and teaching staff need to stress the importance of safe work and encourage students to refresh their knowledge of safety rules regularly. A similar conclusion was drawn by Walters et al. (2017); there is a need to improve awareness of chemical laboratory safety among students; further education and training on this should be part of students’ studies [36].

#### 4.2.2. Employees

In total, work accidents have been reported twice as often as commuting accidents, especially in HEIs mainly offering programs in the engineering and scientific fields. The wide field of engineering, for example, means working with different types of machines, which might increase the risk of work accidents. This profile is consistent with the occurrence of almost a third of all work accidents at the University of Applied Sciences Trier classified as injuries by mechanical impact or by physical, chemical, or biological impact. For the University of Applied Sciences Kaiserslautern, these categories accounted for six out of all seven employees’ work accidents. Our results show that there is a significant association between the person group (employees vs. students) and the type of accident (commuting vs. work), revealing that significantly more employees experienced a work accident.

However, why do employees at the mentioned HEIs seem to be more at risk for work accidents? In the higher education context, it is possible that the theoretical part of learning and teaching sometimes outweighs the practical part; teaching staff might not be able to keep up with practice as much as other employees, who focus more on practice and have a relevant background. This study thus suggests a need to promote regular training and provide information on how to create a safe working environment, which could also serve as a good role model for students striving to work in the same field later in their career.

Employees injured by falling also accounted for a major part of the work accidents in the accident reports of all three HEIs. Falls can result in fractures and other injuries, disability, and fear, and they can trigger a decline in physical function and loss of autonomy [37]. According to the WHO, falls are the second leading cause of unintentional injury-related deaths worldwide [38]. Age is one of the key factors for falls and falls resulting in serious injuries [38]. Therefore, their prevention is important, especially for older employees. The promotion of workplace safety regulations and safety programs needs to be stressed.

### 4.3. Commuting Accidents

#### 4.3.1. Employees and Students

In addition to injuries by falling, the commuting accidents of employees at the German University of Administrative Sciences Speyer, the University of Applied Sciences Trier, and the University of Applied Sciences Kaiserslautern often involved injuries by a bump or hit. More specifically, employees suffered from these accidents by colliding with someone or something. We can see comparable results with the students. The most common causes of commuting accidents for students were injuries by falling and by bumps or hits as well. In addition to that, we were able to reveal students to have a significantly higher risk for experiencing commuting accidents. We can discuss whether students might be showing a less careful behavior in the context of commuting as they feel experienced and secure in the context of cycling or driving (this will be further discussed in the paragraph below). In contrast to that, students might behave more careful in the context of their study work as they are probably not as experienced and therefore more consequently following safety instructions.

#### 4.3.2. Non-Motorized Transport Commuting Options

In our dataset, there was no particular coding of whether the commuting accidents were traffic-related or involved other reasons. A similar study on study-related accidents at the JGU revealed that more than 40% of commuting accidents were cycling related [17]. As this former study focused on student accidents only, this high number of cycling accidents might not be consistent with employees’ accident rates because employees might commute by car more often than students do. However, in the following, we suppose that cycling plays an important role in the prevention of commuting accidents and injuries for both employees and students. A North American review of cycling trends showed that cycling is concentrated in central cities, especially near universities [39]. As our data deal with accident rates at HEIs, this circumstance is probably applicable to our study as well.

Clearly, the prevention of bicycle accidents is important to reduce physical injuries and possibly serious injuries of the individuals concerned. A review of the medical records of patients with cycling-related injuries who were admitted to Sørlandet Hospital in Kristiansand (Norway) between 2012 and 2014 showed, among other things, that fractures and minor head injuries were most frequent. A total of 10% of cycling-related injuries reported were considered severe and critical for the adult group, with four adults showing significant sequelae after 12 months [40]. Such a study presented several starting points for prevention that also apply to the present research. Incidents of head or neck injuries, for example, could be reduced with a change in helmet policy. The positive effects of using bicycle helmets have already been assessed in multiple studies and are not new in the field of public health.

Preventing bicycle accidents is not only necessary to maintain people’s health but also to prevent the subsequent burden of medical costs. A Belgian study about minor accidents by bike (bicycle accidents not involving death or heavily injured individuals and with possible hospital visits lasting less than 24 h) that used several electronic surveys analyzed the economic costs of 118 accidents. On average, EUR 841 was the estimated total economic cost, with 48% of the amount resulting from productivity loss. Furthermore, 27% of the total costs were intangible, representing an important burden related to minor bicycle accidents [41]. Although the preceding section dealt with the negative aspects of bicycle use, it is important to mention its positive effects and to show why, from a public health perspective, recommending bicycle riding and investing in prevention programs remains important. Studies have shown that bicycle riding as a physical activity has major health effects. It plays a role in reducing the risk of many adverse health conditions and increasing ones’ life expectancy. For example, the incidence of major non-communicable diseases, such as coronary heart disease, type 2 diabetes, and even breast and colon cancers, has a direct connection with one’s level of physical activity [42]. Possible prevention programs should therefore implement a change in infrastructure, such as the improvement of bicycle lanes and paths or traffic calming. In addition, bicycle users should have access to safety training programs that raise their awareness of safe bicycle use.

As bicycles are not the only relevant means of transportation, expanding the field of commuting accidents (skateboards and longboards, for example) might be helpful regarding prevention programs within the context of HEIs.

#### 4.3.3. Cars

A probably much larger number of commuting accidents than those caused by skateboards occur by driving. Multiple studies on car accidents involving young drivers, just like our group of students at HEIs, have been conducted. One British retrospective longitudinal study identified the factors contributing to crashes, such as challenging driving conditions, risk-taking behaviors, and inexperience. In particular, the combination of these factors has a major impact on young drivers’ accidents. For example, slippery roads because of poor weather are great risks to young drivers who are inexperienced and are likely to exceed the recommended speed [43].

However, as this study focused on not only students but also employees who were involved in car accidents, more risk factors could be identified in commuting accidents and in the context of HEIs.

### 4.4. Limitations

The most important limitation of this study was the potential lack of reporting. The individuals involved in accidents might not go to the doctor at all after an accident, thinking that it is unnecessary and that they can treat their injuries by themselves. They might not know about the need for doctors to document such accidents and injuries for their cases to be classified as study or work related, as well as about the need for doctors to report the incident to the responsible local accident insurance funds. When at work or when studying at HEIs, employees and students, respectively, are more likely to be in contact with professional support and other staff who know about the reporting of cases. Therefore, we can assume that the accurate number of accidents, especially of commuting-related ones, was underestimated in the present study. Another limitation was the comparison of accident rates between employees and students, as their total numbers differed substantially. Having a look at Appendix A, one can see that at the German University of Administrative Sciences Speyer there are only about 100 more students than employees while at the University of Trier for example there are 6–7 times more students than employees. Considering those rather big differences in between the different HEIs as well, the comparability is limited. Furthermore, because of the use of secondary data, we could only compare frequencies in some aspects. However, where we had access to the distribution of the whole population, we calculated the modified thousand-men quota.

## 5. Conclusions

In the present study, the aim was to address the gap of examination of work and commuting accidents in the context of higher education institutions (HEIs). The complete and objective accident data provided by the Accident Insurance Fund of Rhineland-Palatinate, Germany, helped identifying certain settings creating a higher risk for employees or students to suffer an accident. Sporting accidents are the major cause for work accidents among students, but also needlestick injuries (NSIs) and laboratory accidents (mainly among medical students or students of natural sciences) are important causes of injuries at HEIs, whereas employees could be identified to be at a higher risk when working in the field of engineering. For both parties, traffic-related commuting accidents played an important role. This evidence-based identification of potential risk groups is important as a guide for programs more individualized programs of health promotion at HEIs, making them more effective.

## Figures and Tables

**Figure 1 ijerph-20-02462-f001:**
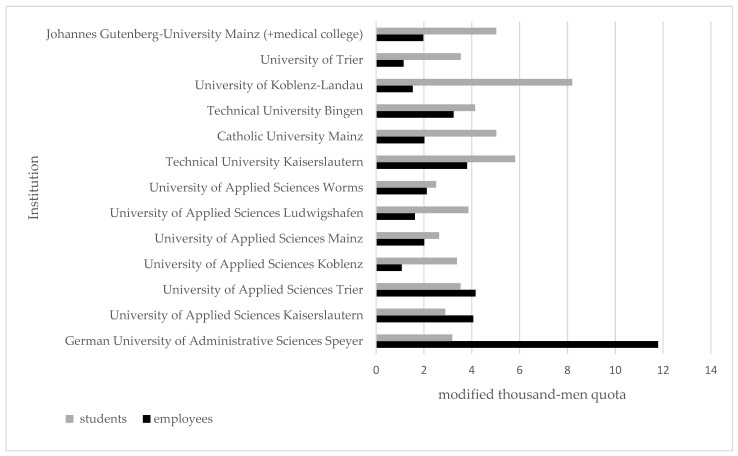
Modified thousand-men quotas (mTMQs) of employees and students from December 2014 to December 2019 for the different institutions. The mTMQ describes the rate of injuries per 1000 employees or 1000 students by adding up the total number of accidents during this period, multiplying this by 1000, and then dividing the result by the average number of employees or students at the institution during the five-year period.

**Table 1 ijerph-20-02462-t001:** Accident data and sample characteristics.

	Total Sample N (%)	Age at the Time of Accident M ± SD	Type of Accident N (%) *	
			Commuting	Work
	3810 (100%)	25.3 ± 7.2	1326 (34.8%)	2484 (65.2%)
**Category of person**				
Employee	426 (11.2%)	37.7 ± 12.8	128 (30.0%)	298 (70.0%)
Student	3384 (88.8%)	23.7 ± 4.0	1198 (35.4%)	2186 (64.6%)
**Gender**				
Female	2021 (53.0%)	25.1 ± 7.3	847 (41.9%)	1174 (58.1%)
Male	1789 (47.0%)	25.4 ± 7.0	479 (26.8%)	1310 (73.2%)

* Percentages as a proportion of the respective columns of the total sample.

**Table 2 ijerph-20-02462-t002:** Characteristics of students’ accidents at the University of Koblenz-Landau.

Institution	Type of Accident	General Reason for the Accident	Specific Reason for the Accident
**University of Koblenz-Landau (N = 665; 100%)**	Work * (N = 403; 61%)	Injured by falling (N = 213; 52.9%)	Twisting one’s ankle (N = 118; 55.4%)
			Falling, tripping (N = 40; 18.8%)
			Failed rotation (N = 29; 13.6%)
			Slipping (N = 20; 9.4%)
			Falling off something (N = 4; 1.9%)
			Failed jump (N = 2; 0.9%)
		Injured by a bump/hit (N = 129; 32.0%)	Getting hit by something (N = 68; 52.7%)
			Ramming something (N = 27; 20.9%)
			Colliding with someone/something (N = 24; 18.6%)
			Getting hit/kicked/scratched by someone (N = 7; 5.4%)
			Getting pushed over by someone/something (N = 2; 1.6%)
			Somersaulting (N = 1; 0.8%)
		Injured by mechanical impact (N = 35; 8.7%)	Cutting damage/stab injuries (N = 12; 34.3%)
			Others (N = 12; 34.3%)
			Squeezing oneself (N = 8; 22.8%)
			Getting bit (N = 3; 8.6%)
		Injured because of incorrect loading (N = 16; 4.0%)	Failed landing (N = 16; 100%)
		Others (N = 6; 1.5%)	Getting caught in something (N = 6; 100%)
		Injured by physical, chemical, or biological impact (N = 4; 0.9%)	Damage through hazardous substances (N = 3; 75.0%)
			Burning oneself (N = 1; 25.0%)
	Commuting N = 262 (39%)	Injured by a bump/hit (N = 163; 62.2%)	Colliding with someone/something (N = 142; 87.1%)
			Ramming something (N = 14; 8.6%)
			Getting hit by something (N = 3; 1.8%)
			Somersaulting (N = 3; 1.8%)
			Getting hit/kicked/scratched by someone (N = 1; 0.6%)
		Injured by falling (N = 93; 35.6%)	Falling, tripping (N = 48; 51.6%)
			Slipping (N = 25; 26.9%)
			Twisting one’s ankle (N = 18; 19.4%)
			Falling off something (N = 1; 1.1%)
			Failed rotation (N = 1; 1.1%)
		Injured by mechanical impact (N = 3; 1.1%)	Squeezing oneself (N = 3; 100%)
		Others (N = 3; 1.1%)	Getting caught in something (N = 3; 100%)

* Significant difference between the number of work and commuting accidents in Chi²-Test using a 1:1 ratio as expected values, *p* < 0.001.

**Table 3 ijerph-20-02462-t003:** Characteristics of students’ accidents at Technical University Kaiserslautern.

Institution	Type of Accident	General Reason for the Accident	Specific Reason for the Accident
**Technical University Kaiserslautern** **(N = 406; 100%)**	Work * (N = 321; 79%)	Injured by falling (N = 138; 43.0%)	Twisting one’s ankle (N = 66; 47.8%)
			Falling, tripping (N = 49; 35.5%)
			Failed rotation (N = 12; 8.7%)
			Slipping (N = 6; 4.3%)
			Failed jump (N = 3; 2.2%)
			Falling off something (N = 2; 1.5%)
		Injured by a bump/hit (N = 76; 23.7%)	Getting hit by something (N = 32; 42.1%)
			Ramming something (N = 15; 19.7%)
			Colliding with someone/something (N = 15; 19.7%)
			Getting hit/kicked/scratched by someone (N = 9; 11.8%)
			Getting pushed over by someone/something (N = 4; 5.3%)
			Somersaulting (N = 1; 1.3%)
		Injured by mechanical impact (N = 66; 20.6%)	Cutting damage/stab injuries (N = 42; 63.7%)
			Squeezing oneself (N = 7; 10.6%)
			Others (N = 17; 25.8%)
		Injured by physical/chemical/biological impact (N = 30; 9.3%)	Damage through hazardous substances (N = 26; 86.7%)
			Burning oneself (N = 2; 6.7%)
			Damage through disease vectors (N = 1; 3.3%)
			Damage through electricity (N = 1; 3.3%)
		Injured because of incorrect loading (N = 7; 2.2%)	Failed landing (N = 7; 100%)
		Others (N = 4; 1.2%)	Getting caught in something (N = 4; 100%)
	Commuting (N = 85; 21%)	Injured by falling (N = 42; 49.4%)	Falling, tripping (N = 28; 66.7%)
			Twisting one’s ankle (N = 9; 21.4%)
			Slipping (N = 5; 11.9%)
		Injured by a bump/hit (N = 43; 50.6%)	Colliding with someone/something (N = 34; 79.1%)
			Ramming something (N = 5; 11.6%)
			Getting pushed over (N = 1; 2.3%)
			Getting hit/kicked/scratched by someone (N = 1; 2.3%)
			Getting hit by something (N = 1; 2.3%)
			Somersaulting (N = 1; 2.3%)

* Significant difference between the number of work and commuting accidents in Chi²-Test using a 1:1 ratio as expected values, *p* < 0.001.

**Table 4 ijerph-20-02462-t004:** Characteristics of students’ accidents at Johannes Gutenberg University Mainz.

Institution	Type of Accident	General Reason for the Accident	Specific Reason for the Accident
**Johannes Gutenberg University Mainz ** **(N = 771; 100%)**	Work * (N = 660; 85.6%)	Injured by falling (N = 279; 42.3%)	Twisting one’s ankle (N = 132; 47.3%)
			Falling, tripping (N = 59; 21.1%)
			Failed rotation (N = 58; 20.8%)
			Slipping (N = 15; 5.4%)
			Failed jump (N = 8; 2.9%)
			Falling off something (N = 7; 2.5%)
		Injured by mechanical impact (N = 151; 22.9%)	Cutting damage/stab injuries (N = 122; 80.8%)
			Others (N = 24; 15.9%)
			Squeezing oneself (N = 2; 1.3%)
			Getting bit (N = 2; 1.3%)
			Damaged by an explosion (N = 1; 0.7%)
		Injured by a bump/hit (N = 148; 22.4%)	Getting hit by something (N = 71; 48.0%)
			Colliding with someone/something (N = 40; 27.0%)
			Ramming something (N = 28; 18.9%)
			Getting hit/kicked/scratched by someone (N = 6; 4.1%)
			Somersaulting (N = 2; 1.4%)
			Getting pushed over (N = 1; 0.7%)
		Injured by physical/chemical/biological impact (N = 59; 8.9%)	Damage through hazardous substances (N = 35; 59.4%)
			Damage through disease vectors (N = 13; 22.0%)
			Burning oneself (N = 11; 18.6%)
		Others (N = 12; 1.8%)	Getting caught in something (N = 11; 91.7%)
			Strangulation (N = 1; 8.3%)
		Injured because of incorrect loading (N = 11; 1.7%)	Failed landing (N = 10; 90.9%)
			Failed lifting (N = 1; 9.1%)
	Commuting N = 111 (14.4%)	Injured by falling (N = 65; 58.6%)	Falling, tripping (N = 31; 47.7%)
			Slipping (N = 18; 27.7%)
			Twisting one’s ankle (N = 15; 23.1%)
			Failed rotation (N = 1; 1.5%)
		Injured by a bump/hit (N = 45; 40.5%)	Colliding with someone/something (N = 36; 80.0%)
			Ramming something (N = 5; 11.1%)
			Somersaulting (N = 4; 8.9%)
		Injured by mechanical impact (N = 1; 0.9%)	Cutting damage/stab injuries (N = 1; 100%)

* Significant difference between the number of work and commuting accidents in Chi²-Test using a 1:1 ratio as expected values, *p* < 0.001.

**Table 5 ijerph-20-02462-t005:** Characteristics of employees’ accidents for the three institutions with the highest modified thousand-men quotas.

Institution	Type of Accident	General Reason for the Accident	Specific Reason for the Accident
**German University of Administrative Sciences** **Speyer** **(N = 12; 100%)**	Work ^ns^ (N = 7; 58.3%)	Injured by falling (N = 5; 71.4%)	Falling, tripping (N = 2; 40.0%)
			Twisting one’s ankle (N = 2; 40.0%)
			Slipping (N = 1; 20.0%)
		Injured by mechanical impact (N = 1; 14.3%)	Squeezing oneself (N = 1; 100%)
		Injured by physical/chemical/biological impact (N = 1; 14.3%)	Burning oneself (N = 1; 100%)
	Commuting (N = 5; 41.7%)	Injured by falling (N = 4; 80.0%)	Falling, tripping (N = 2; 50.0%)
			Slipping (N = 1; 25.0%)
			Falling off something (N = 1; 25.0%)
		Injured by a bump/hit (N = 1; 20.0%)	Colliding with someone/something (N = 1; 100%)
**University of Applied Sciences Trier** **(N = 18; 100%)**	Work ^ns^ (N = 13; 72.2%)	Injured by a bump/hit (N = 6; 46.2%)	Getting hit by something (N = 4; 66.7%)
			Colliding with someone/something (N = 1; 16.7%)
			Ramming something (N = 1; 16.7%)
		Injured by falling (N = 3; 23.1%)	Falling, tripping (N = 2; 66.7%)
			Slipping (N = 1; 33.3%)
		Injured by mechanical impact (N = 3; 23.1%)	Cutting damage/stab injuries (N = 2; 66.7%)
			Others (N = 1; 33.3%)
		Injured by physical/chemical/biological impact (N = 1; 7.6%)	Damaged by radiation (N = 1; 100%)
	Commuting (N = 5; 27.8%)	Injured by a bump/hit (N = 4; 80.0%)	Colliding with someone/something (N = 4; 100%)
		Injured by falling (N = 1; 20.0%)	Falling, tripping (N = 1; 100%)
**University of Applied Sciences Kaiserslautern ** **(N = 13; 100%)**	Work ^ns^ (N = 7; 53.8%)	Injured by physical/chemical/biological impact (N = 3; 42.9%)	Damage through hazardous substances (N = 3; 100%)
		Injured by mechanical impact (N = 3; 42.9%)	Cutting damage/stab injuries (N = 1; 33.3%)
			Squeezing oneself (N = 1; 33.3%)
			Others (N = 1; 33.3%)
		Injured by falling (N = 1; 14.2%)	Twisting one’s ankle (N = 1; 100%)
	Commuting (N = 6; 46.2%)	Injured by falling (N = 3; 50.0%)	Failed rotation (N = 1; 33.3%)
			Twisting one’s ankle (N = 1; 33.3%)
			Slipping (N = 1; 33.3%)
		Injured by a bump/hit (N = 3; 50.0%)	Colliding with someone/something (N = 3; 100%)

^ns^ non-significant difference between the number of work and commuting accidents, *p* = 0.05 level, in Chi²-Test using a 1:1 ratio as expected values.

## Data Availability

The raw data supporting the conclusions of this article will be made available by the authors without undue reservation.

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
