# Peer review of "Differences in Work and Commuting Accidents between Employees and Students at Higher Education Institutions in Rhineland-Palatinate, Germany, from December 2014 to December 2019"

_ijerph, 2023, doi:10.3390/ijerph20032462_

Round 1
Reviewer 1 Report
This paper reports on study-related accidents in students and employees at higher education institutions in Rhineland-Palatinate, Germany. The study provides data on an important public health crisis and fills a gap in the injury literature. The paper is well-written and includes detailed analyses. However, there are major concerns related to clarity, organization, and presentation of the results.
Major Comments
Abstract
· “modified thousand-man quotas”
· “The frequency and 21 characteristics of the accidents varied between employees and students, as well as between 22 institutions.”
· Well-written, clear, strong abstract
Introduction
· The first paragraph of the introduction needs to clarify if the paper is focusing on motor vehicle accidents or all accidents. More specifically, the first sentence sounds as though the authors are referring to all kinds of accidents but then the second two sentences and remaining paragraphs sound more like motor vehicle related accidents.
· I am unsure if it is accurate and/or clear to describe the definition of the accident with the type of person involved. For example, on page 1, lines 34-35, the authors write: “it is important to mention that the definition of accidents here includes not only employees and their professional activities but also higher education students.” I do not think this is as relevant to the actual definition of the term “accident.” Is it possible that the authors instead are conceptualizing the purpose of the paper to include both employees and students? If so, the language may need to simply be altered slightly to reflect this intent.
· The introduction contains strong content and support for the purpose of the paper, however, it could benefit from some reorganization within part 1.1 specifically. For example, much of the content in the paragraph included on page 2, starting line 45, could better be placed at the beginning of the introduction as it discusses accidents broadly. Then, the introduction could narrow in toward German accidents, and the type of accidents specific to their study.
· I appreciate the section on public health relevance of accidents and feel it provides strong support for the impact of your research.
· This may be a cultural barrier, but I am a little confused about the role of HEIs for this paper. It sounds as though they are a place for work, a place for study, but also a place to promote healthy life choices? Is this referring to HEIs promoting healthy life choices for the community (not at the HEI) or is it referring to promoting these choices within the employees and students at the HEI?
· Based on the paragraph starting on page 2, line 98, it sounds as though the paper may actually be focusing on all accidents. If this is the case, further clarity is needed in the abstract and introduction to clarify the types of accidents focused on for the present paper and why we are emphasizing commuting in the abstract.
· The paragraphs that focus on injury rates within HEI are the most interesting and relevant to the present study. The authors should consider condensing some of the earlier paragraphs on injury less relevant to the present study to help clarify the paper’s focus and to make the introduction more concise.
· The aims and research questions paragraph is helpful and sets up the paper well.
Method
· This section was well-written, clear, and easy to follow.
· I am unfamiliar with the modified thousand-man quota (mTMQ) methodology, please refer to a reviewer with expertise in this area for feedback.
Results
· Is “getting hit, kicked, or scratched” always accidental/unintentional? Was that accounted for in data collection? (page 7, 310 and 316)
· What does “somersaulting” mean in this context? (page 7, 311 and 315)
· The results are overall well-written, but each paragraph is very dense with descriptive statistics. I would recommend condensing each paragraph to mention highlights and key findings. Then, the tables provide all of the same information and can be referred to for more details.
Discussion
· Overall, the discussion does a good job of summarizing the results, providing interpretation, and offering ideas for intervention.
· The phrase “study- related accidents” is a great phrase to capture the types of injuries examined for this paper. It may be better to include that at the very beginning to set up the paper rather than referring to the injuries as “work” or “commuter” related.
· No formal comparisons were made between the two groups so just be careful with the language used when describing the examination of the data.
· I am not sure what is meant by the first two sentences of 4.1: lines 405-407. Can this be re-written/clarified?
· This sentence was also hard to follow and can be re-written: “Students accounted for a larger number of the reported accidents, as a look at the total numbers shows that there were slightly (only 3.0%–3.5% more) more women than men involved in study-related accidents.” (page 12, line 411)
· Given the imbalance in gender and imbalance in employees versus students. Are there any other thoughts or considerations for how the data could have been examined or analyzed rather than dismissing the findings as inconsequential?
· The finding that women are at greater risk is actually different than many other studies, especially in children. The authors should comment on why these findings may be different in this setting and in older age groups.
· I am not sure if I agree with the conclusion that more mature or older students may be at less risk for study-related accidents due to more life and work experiences. What is the reasoning for this rationale? Is there specific evidence that can be cited to support this claim?
· Was their confirmation that many of the accidents were in sports program or is it an assumption that is being made? If it is an assumption, that needs to be more clear.
· It seems as though the only form of commuting was bicycling. Is that true? Overall, the commuting paragraphs appear to be the most confusing at this time. It would be beneficial if the authors could clarify what is meant by commuting, what modes of transportation are actually being captured and collected in the data, and what can be done in the future to address this topic.
· What is meant by a limitation being a comparison of accident rates between employees and students? This was not clearly illustrated in the results or discussion.
Tables/Figures
· Figure 1 is overall presented well and easy to read. There is a label “modified mean injury rate” that seems randomly placed but it may be the label for the x-axis. Perhaps it would be easier to recognize if it is closer to the axis line.
· All of the tables are helpful and well-formatted.
Minor Comments:
· On page 2, lines 48-49, the sentence reads: “Accident and injury statistics [4] focus on four aspects: deaths from accidents and assault, the extent of accidents, and health care for injuries.” I am only counting three aspects here.
· Page 2, line 74 is the first time you refer to accidents as “unintentional injuries” it may be helpful to say that earlier so that unfamiliar readers are not confused.
· Typo, page 7, line 306: “falling of something (n = 4; 1.9%)”
· What does “Failed rotation” mean? (page 7, line 306)
· “Lab work” is a slightly misleading title – maybe “Lab-related Injuries” or “Injuries in a Lab Setting”
Reviewer 2 Report
Thank you for the opportunity to reviewer your paper, given my training, practice and research/publications experience related to injuries and illness via accidents (mostly preventable incidents) in U.S. among students and adults (teachers, administrators, other staff) in career-technical education and work-based learning. This is well-written paper, with detailed tables. Please see attached files for comments to address/edits to make. I am having trouble attaching to files, so, here is the one major comment about your Supplemental file containing Tables S1, S2 and S3:
"Every Table must stand alone, including if in an online Supplement. The Tables S1 and S2 are good/ready. The Table S3, however, requires edits/additions. Please add either a second sentence to the title, or footnote to each of the two highlighted columns, to specify the complete definition of the mean injury rate of ________ (employees, or, students). Is it mean of annual rates per 1000 ________ (employees, or, students)? In addition, the mean injury rate data should only be to one decimal place to be consistent with other columns. Finally, and this is optional but recommended, this reviewer believes the "Mean number of _____, December 2014 to December 2019" columns (for employees, and, for students) should be to nearest whole number. Why decimal place/fraction of a person? Not necessary, even if math correct."

Reviewer 3 Report
Thanks for the opportunity to read your paper. Unfortunately, the manuscript reads like a general technical report, possibly a part of a Bachelor thesis. The lack of statistical treatment of your data and respective emerging findings limits the scientific value of the article.
I encourage you to process your data further and explore what it comes out from the analysis. Please see below my comments on several parts of your manuscript.
Abstract
-Avoid using the term "investigate" for your study as it has a specific meaning for accidents/incidents.
-The main findings are not reported.
Mina body
-Lines 43-44: Citation needed
-Lines 50-51: Why do you name the 3.2% portion of all deaths as one of the most common causes?
-Lines 50-58: This part can be removed. I cannot see what value it adds.
-Lines 58-65: Why are all those figures reported here? You can decrease this part significantly and be more concise. Merge with Lines 66-69.
-Line 87: Why "collide"?
-Lines 105-106: What portion of those were in the study context?
-Section 1.3: A Table summarising each of the studies reviewed for student populations would be great.
-Section 2.3
--Why only descriptive statistics and not some basic statistical tests to explore some first associations between the variables you included in the study? This will then allow you to discuss any (non) significant results against studies in the same or other sectors with similar variables. Now, the paper reads more like a simple technical report rather than a scientific piece of work with analyses that offer insights.
--In general, you need to consider what variables you can compute further to increase the validity of your comparisons and tests. For instance, you cannot really compare rates/frequencies of accidents occurring at work or during commuting if you do not correct for and consider the relative rate of exposure/population representation (e.g., 2 hours commute time on average vs 8 hours work in average?). Another example: what is the male/female distribution in the target population when you compare gender-clustered results? You must apply whatever corrections you can reasonably perform from the data you have.
---There is no exposure data in the study for any of the variables (e.g., how many students commute by what means and for how long). You cannot claim anything about populations at risk. You can only observe and suggest where the focus could initially be.
-Sections 3.1 & 3.2
--No reason to report in the text so many figures since you already present them in the tables. Focus on the most/least frequent cases or anything else deserving attention.
--The reference to each of the universities with so much detail does not add any value. You can move the descriptive results to the supplementary file and report here the results from statistical tests, as suggested above.
-Section 4:
--This needs to be enriched and modified based on statistical test results. Expectedly, most of the text in this section will change. The authors now refer to the figures more from a qualitative perspective.
--For instance, in lines 405-408, the accidents can be more for students but the rates are higher for employees. You must compare similar and comparable entities. You again wrongly compare only frequencies in section 4.2.
--The sections referring to various characteristics/results based in frequency (e.g., types of accidents of students or commuting accidents) refer to literature but the discussion needs to elevate to an academic level and minimise normative language, especially about prevention as (1) this was not within the scope of this study, and (2) prevention needs to happen within each context.
Round 2
Reviewer 1 Report
This paper has sufficiently been revised to meet the feedback I provided. My main remaining concern is a careful revise of the new text to ensure all added sections are succinct and clear. Often, the added text was too wordy and could be cut down. A point of clarity, however, with my concern regarding the motor vehicle accidents versus the general accidents regards the words “commuter” versus “work.” These labels were misleading to me personally, but I think the authors provide sufficient evidence to retain the language they use.
Reviewer 3 Report
Thanks for considering my comments and for revising your article. Unfortunately, the main remark about statistics was not seriously considered. I do not know what made you think I would change my perspective about your data treatment and why you did not already consult with a statistician.
Indeed, your dataset does not allow any sophisticated statistical tests, but you can still perform some and reveal further insights. I am only trying to help you produce something that is more than descriptive and probably more informative.
Please consider the following:
-Some ideas for statistics (mainly between categorical variables - Chi-square or Fisher's Exact tests). You can think of more, but make sure they "make sense" Just mind the accumulation of type error I and apply the Bonferroni correction.
--Category vs HEI
--Gender vs HEI, controlled for Category
--Accident type vs HEI, controlled for Category
--General reason vs HEI, controlled for Accident type and Category
--Specific reason vs HEI, controlled for Accident type and Category (if the sample distribution allows this detail)
--Gender vs Accident type, controlled for Category
--Gender vs General reason controlled for Accident type and Category
-Figure 1 does not show the names of all HEI - please modify
-No justification of this "Furthermore, we examined the accident characteristics of the three institutions with the highest modified mean injury rates".
-Your response to comment No 17 on the previous version: I cannot understand your rationale. Since, for example, you have numbers of students and types of accidents per HEI, you can still refer to rates and not frequencies (e.g., in section 4.2). You just have to do the maths.
-The Discussion section is unjustifiably long as it presents information that is not always relevant or traceable to the findings of your study. You often assume things you cannot find in your data and then discuss other studies without any reason. You must discuss only literature relevant to what you studied and found. For example, section 4.3.2 adds no practical value since you have no data to compare and discuss.
-I expect that the (non) significant results from the statistics will give you more material and grounds for the Discussion section.
